# Cerebral Hemodynamic Changes during Unaffected Handgrip Exercises in Stroke Patients: An fNIRS Study

**DOI:** 10.3390/brainsci13010141

**Published:** 2023-01-13

**Authors:** Yuqin Ma, Yang Yu, Wen Gao, Yongfeng Hong, Xianshan Shen

**Affiliations:** 1Department of Rehabilitation Medicine, The Second Hospital of Anhui Medical University, Hefei 230601, China; 2Department of Rehabilitation Medicine, Second Clinical Medical College, Anhui Medical University, Hefei 230031, China

**Keywords:** functional near-infrared spectroscopy (fNIRS), ipsilesion, stroke, cross education, lateral index (LI)

## Abstract

This study aimed to assess the effect of the altered strength of the sound limb on the hemodynamics in the affected brain of stroke patients. We recruited 20 stroke patients to detect changes in the HbO concentrations in the bilateral prefrontal cortex (PFC), sensorimotor cortex (SMC), and occipital lobe (OL). We performed functional near-infrared spectroscopy (fNIRS) to detect changes in oxyhemoglobin (HbO) concentrations in regions of interest (ROIs) in the bilateral cerebral hemispheres of stroke patients while they performed 20%, 50%, and 80% maximal voluntary contraction (MVC) levels of handgrip tasks with the unaffected hands. The results suggest that when patients performed handgrip tasks with 50% of the MVC force, SMC in the affected cerebral hemisphere was strongly activated and the change in the HbO concentration was similar to that of the handgrip with 80% of MVC. When the force was 50% of MVC, the SMC in the affected hemisphere showed a more proportional activation than that at 80% MVC. Overall, this research suggests that stroke patients with a poor upper limb function should perform motor training with their sound hands at 50% of the MVC grip task to activate the ipsilesional hemisphere.

## 1. Introduction

Ischemic stroke has a high incidence of disease and disability [1]. After a stroke, patients often show different degrees of limb hemiplegia. Symptoms include, but are not limited to, muscle weakness, paresthesia, and decreased coordination [2], which makes it difficult for patients to return to their families and society. The interruption of the cerebral arterial blood flow due to multiple causes is fundamentally why patients develop a limb impairment. In such cases, neurons in the brain area that innervate the affected limb develop ischemia and hypoxic necrosis, leading to irreversible neuronal damage [3]. The cerebral cortex of healthy people exerts interhemispheric interactions through intrinsic connections [4], but these connections will change after stroke. During the acute stage of stroke, the motor function is associated with the preservation of the corticospinal tract. The subsequent recovery process is related to the connection of the brain network between the two hemispheres [5]. Therefore, the basis of limb rehabilitation for patients involves promoting the functional compensation of brain nerve cells around the infarct foci. The degree of neurological remodeling directly affects the patient’s potential for motor recovery during convalescence [6].

When a healthy person performs a fine unilateral limb movement, one cerebral hemisphere activates, and then suppresses the activation of the contralateral hemisphere through the corpus callosum. This allows for the more precise control of the moving limb [7]. However, the interhemispheric inhibition changes in patients with stroke. The interhemispheric competition model suggests that a patient’s contralesional hemisphere inhibits the activation of the affected hemisphere [8]. Meanwhile, the contralesional hemisphere showed an overactivation because of the relatively lower inhibition from the affected hemisphere. In other words, the hypermobilization of the unaffected limbs hinders the recovery of the affected limbs. However, several studies have shown that stimulating the unaffected hemisphere is beneficial for the recovery of the function of the affected side [9,10]. In addition, this model cannot explain the fact that when the corpus callosum is damaged, the inhibition of the contralesional hemisphere on the affected side should also be reduced; however, patients with a damaged corpus callosum generally exhibit a worse upper limb function on the affected side [11]. Indeed, the previous model of interhemispheric inhibition was not applicable to all patients. There is neither a specific interhemispheric inhibition nor a specific interhemispheric excitation between the two cerebral hemispheres, but rather the inhibition and excitation act simultaneously between the two hemispheres; this phenomenon is termed “crossed surround inhibition” [12]. In addition, D Bundy et al. [13] and M Aswendt et al. [14] found that in function studies in stroke patients, the activation of the contralesional hemisphere plays a vital role in recovery. This is because residual network connectivity in the healthy hemisphere replaces the function of the damaged area. This is known as the vicariation model. Within 6 months after stroke, the primary motor cortex region (M1) of the affected hemisphere is positively correlated with the degree of functional connectivity of the thalamus, the supplementary motor area, and middle frontal gyrus of the contralesional hemisphere, as well as the degree of recovery of the limb motor function [15]. Reducing the excitability of the contralesional hemisphere with a non-invasive treatment will reduce the compensatory effect, thereby influencing the recovery of the function in the affected hemisphere. Although the function of the corpus callosum and the role of the interhemispheric inhibition model in the rehabilitation of stroke patients still needs to be further studied, the effect of the contralesional hemisphere on the affected side cannot be ignored.

Approximately 73% to 88% of patients with first-time stroke develop upper extremity dysfunction [16]. As such, there is an urgent need to improve the function of the upper limbs of the hemiplegic hand [17]. In patients with severe motor deficits, many rehabilitation treatments are not applicable and have little effect [18], while cross-education has been proven to be an effective means of an adjuvant rehabilitation in many studies [19,20,21]. Cross education is a form of rehabilitation treatment in which the ability of both limbs is increased through training on one side. This approach is indicated for patients who are unable to exercise one limb, such as those with an acute limb injury, postoperative limb fixation, and certain neurological disorders in which the unilateral muscle weakness predominates [22]. This is particularly effective in the rehabilitation of post-stroke patients [19,23]. Similar studies have discovered that in rehabilitation, the training of one limb can increase the strength of the contralateral limb to 7.8% [24]. In addition, it has also been reported that after the strength training of the unaffected limb, the disuse atrophy of the muscles of the affected limb immobilized by a cast can be prevented [25]. Although the theoretical mechanism of cross education has not yet been clearly studied, some related studies have indicated that the activation of the ipsilateral hemisphere may be caused by the co-action of corticospinal fibers [26], contralateral corticospinal fibers [27], and the corpus callosum [28] after the activation of the contralateral hemisphere. Other scholars believe that this is caused by the direct effect on Ia inhibitory neurons or the actions of Renshaw cells on the opposite homologous motor neurons [22]. The cross-education theory undoubtedly provides us with a new rehabilitation strategy for patients with unilateral skeletal muscle and nerve function impairment.

Functional near-infrared spectroscopy (fNIRS), a non-invasive tool for measuring the brain activation, has been widely used to measure cerebral cortical excitability in healthy individuals and patients with various neuropsychiatric diseases [29]. Compared to functional magnetic resonance imaging (fMRI), fNIRS is more convenient and safe [30]. In recent years, with the rapid development of fNIRS technology, an increasing number of researchers have used this technology to study the relationship between limb strength changes and ipsilateral cerebral hemisphere hemodynamics. Kenichi Shibuya [31] et al. used fNIRS to observe the change in the HbO concentration in the M1 region of both hemispheres when performing the maximal voluntary contraction (MVC) tasks with one hand, and the results showed that the HbO concentration in the M1 region of the ipsilateral hemisphere of the moving limb increased with an increase in force, but the HbO concentration in the M1 region of the contralateral hemisphere did not increase with the increase in force when the force reached a certain level, which indicates that when the contralateral hemisphere is activated to a certain level, the ipsilateral hemisphere plays a role in compensating for the excitability of the contralateral cortex [31]. With an increase in force, in addition to the M1 area, the rest of the brain area on the ipsilateral side of the moving limb is also activated. In one study, Derosière G [26] proposed that when performing handgrip training with different MVC forces, 50% MVC could allow the ipsilateral prefrontal cortex (PFC) to be more activated than the contralateral PFC.

Overall, the above experiments have proven that when one limb moves, it is not only one cerebral hemisphere that functions, but the result of the joint action of both cerebral hemispheres. After stroke, the balance between the bilateral hemispheres of patients is broken, and the degree of damage to the hemispheres differs [32]. The study of laterality dominance between the bilateral hemispheres is more conducive to providing different brain stimulation protocols for a subsequent treatment, thereby improving the patient prognosis. Previous studies [33,34] used the laterality index to represent hemispheric dominance. Therefore, in this study, we calculated the laterality index using HbO data to probe into the laterality dominance. In addition, grasp is the most important function of the hand and a key function of the limb that requires restoration after stroke. Executing hand grasp training is necessary to enhance the hand function after a stroke. However, there is no exact explanation in the clinic of how much force can be used to favor patient recovery. Dai et al. [35] showed the existence of a symmetrical activation in the SM1 region of both cerebral hemispheres when the maximum grip strength is 20%, while Derosière et al. [26] argued that an activation in the SM1 region of the bilateral hemisphere is meaningless when the force is between 5% and 50% of the MVC. Therefore, it is important to explore which level of handgrip strength is more suitable for patients undergoing rehabilitation. We used 50% of MVC as the dividing line and selected 20% MVC as the smaller force and 80% of MVC as the relatively greater force at the front and back equidistance. We used fNIRS to detect the real-time changes in HbO in the regions of interest (ROIs) of stroke patients with a poor limb function on the affected side when performing different levels of grip strength training with their sound hand and evaluate the effect of changing the strength of the unaffected limb on the hemodynamics of the affected cerebral hemisphere to further explore the validity of different force levels in cross-education theory. We hypothesized that the greater the grip strength of the unaffected hand, the higher the HbO value in each ROI, and the more conducive it is to the patient’s recovery. We will further discuss whether the cross-education theory is applicable to the cerebral cortex and other areas other than the sensorimotor cortex (SMC) in order to provide new ideas for the future rehabilitation of speech, consciousness, and other disorders caused by injuries to the frontotemporal lobe or other regions

## 2. Materials and Methods

### 2.1. Participants

For the present study, we enrolled 20 stroke patients hospitalized in the Department of Rehabilitation Medicine of the Second Affiliated Hospital of Anhui Medical University from January 2022 to September 2022 to undergo rehabilitation. Before the start of the experiment, we assessed all the subjects’ cognitive level and muscle strength on the affected side, according to the Fugl–Meyer assessment of their limb function. In addition, all participants were informed of the objects, risks, procedures, and significance of the experiment and signed an informed consent form prior to the initiation of the study. This study was conducted in accordance with the Declaration of Helsinki and approved by the Ethics Committee of the Second Affiliated Hospital of Anhui Medical University (approval number YX2020-103F1).

The inclusion criteria were as follows:Aged between 40 and 70, no gender limitation;Convalescent patients with ischemic stroke (first stroke), and course of disease between 1 and 6 months;Patients with hemiplegia and the muscle strength of both upper and lower limbs on the hemiplegic side ≤ grade 3 (manual muscle testing MMT), without any functional impairment on the healthy side;The ability to perform some aerobic rehabilitation by permission of the attending physician;A normal cognitive level (Mini-Mental State Examination, MMSE score > 21 points), the ability to understand the movement instructions of the rehabilitation therapist;Not taking any psychotropic drugs.The exclusion criteria were as follows:Patients whose condition continued to deteriorate, or whose vital signs were unstable;Brain damage not limited to one hemisphere;Other serious medical conditions, such as having a pacemaker;The patient or their family members refused to sign informed consent.

### 2.2. Experimental Procedure

#### 2.2.1. Basic Data Collection

Basic information about the participants, including their sex, age, site of infarction, and risk factors, was recorded prior to their enrollment. In addition, the researcher also assessed the upper limb muscle strength on the hemiplegic side (MMT manual muscle strength assessment), cognition (MMSE scale), and the MVC of the unaffected hand (grip dynamometer) and calculated 20%, 50%, and 80% of the MVC.

#### 2.2.2. Introduction of fNIRS

In this study, relevant signals of the brain functional activity were acquired using a multi-channel fNIRS system (Nirscan-6000A, Danyang Huichuang Medical Equipment Co., Ltd., China) at a sampling rate of 10 Hz [36]. The wavelengths used were 730, 808, and 850 nm. A total of 32 probes (a between-probe distance of 30 mm), including 16 transmitters and 16 detectors, were designed according to the 10/20 international standard wire system, which together formed 34 channels. Fourteen of the 34 channels were positioned over the bilateral prefrontal cortex (PFC), 18 over the sensorimotor cortex (SMC), and 2 over the occipital lobe (OL) to continuously detect any changes in the HbO concentration in the corresponding channel. (The probe layout is shown in Figure 1)

#### 2.2.3. fNIRS Data Acquisition

All subjects were required to have adequate sleep, maintain a good mental state, and to avoid smoking, drinking alcohol and coffee, taking psychotropic drugs, and performing strenuous exercise before the study. All participants learned to use a handgrip dynamometer before the study and were able to skillfully control their grip strength within the corresponding level. The entire process of the experiment was conducted in a quiet room and the participants sat in a comfortable chair. The participants were required to wear an fNIRS collection cap at the external auditory canal and the apex of the head as reference points, and then place their affected hand on their thigh, while the unaffected hand held the handgrip dynamometers. In addition, participants were asked to relax the whole-body muscles before starting, to keep all body parts still, and to keep their eyes on the grip strength device to ensure that the grip strength was controlled to a predetermined standard when performing the task. All the subjects completed three separate grip strength tasks and each experiment was spaced 10 min apart to ensure that the previous grip task had no effect on the subsequent task. The target levels of force were set at 20%, 50%, and 80% of the MVC. Each subject underwent an experimental block paradigm design, which contains a 10 s steady-hold contraction task followed by 15 s of rest. When the task was finished or the rest time was over, the scheduled program sounded to alert the patients. The conditions of the block paradigm were repeated five times for a total of 125 s. At the end of the task or rest time, the scheduled procedure issued instructions to alert patients. When patients hear the first command, the task begins, and they use their unaffected hand to grasp the grip force, quickly bringing the grip strength value to the designated criteria and maintaining it until the second sound. When the second cue sounded, the patients relaxed completely. This procedure was then repeated. We used an fNIRS device to collect the relative oxygenated hemoglobin concentrations of all channels during the task and rest states.

### 2.3. Data Processing and Analysis

We preprocessed the data collected using fNIRS in the preprocessing module of the NirSpark software. First, the standard deviation of the threshold was set to 6.0, and the amplitude of the threshold was set to 0.5. The movement of the standard deviation combined with cubic spline interpolation was used to remove the motion artifacts [37]. Subsequently, the interference signals caused by the heart rate, breathing rate, and Mayer waves were removed with 0.01 to 0.1 Hz band-pass filtering [38]. In the last step, we set all the differential path-length factors to 6.0 and calculated the Delta HbO concentration of each channel in the rest/task state according to the modified Beer–Lambert law.

The NirSpark software was used to calculate the mean values of the task and rest states in the block mode for analysis. First the block time range was set to [0 s, 25 s] at the start of the task (including a 10 s handgrip and a 15 s rest). The baseline was set to [−2 s, 0 s], and the data from the five blocks were superimposed and averaged to obtain the mean oxygenated hemoglobin values for each channel in the task and rest states. The HbO for each ROI was calculated as follows:The fNIRS data of 10 patients with a left-sided hemisphere infarction were flipped using the mirror transformation technique [39], such that the fNIRS data of all 20 patients with a hemisphere infarction were transformed into a right-sided hemisphere infarction to facilitate data processing.Based on the concentration of oxygenated hemoglobin in all channels in each ROI divided by the number of channels in each ROI, we obtained the mean oxyhemoglobin of each ROI in the task and rest states. The Shapiro–Wilk method was used to test the normality of the differences between these groups, and data corresponding to the normal distribution were expressed as the mean ± standard deviation. The concentration of HbO in the task and rest state of each grip strength level group were tested using a paired *t*-test to detect the activation of the ROIs in each hemisphere.Levene’s test was applied to calculate the homogeneity of variance in each group. A one-way ANOVA and LSD tests were used to determine whether there were differences in the activation of the ROIs between the different handgrip groups.The calculation of the laterality index (LI) was performed as follows: we calculated the average change in ΔHbO (mean of task HbO concentration–mean of rest HbO concentration) in the ROI of each subject’s ipsilesional and contralesional hemispheres using the formula: (mean ΔHbO ipsilesional–mean ΔHbO contralesional)/(mean ΔHbO ipsilesional + mean ΔHbO contralesional) [40,41]. The results were the relative changes in the HbO in the ROIs in the ipsilesional and contralesional hemispheres. If the LI > 0, the ROIs of the ipsilesional hemispheres were dominant. If LI < 0, the ROIs of the contralesional hemisphere were considered dominant.

## 3. Results

### 3.1. Clinical Details of Stroke Patients

Twenty participants in this study were hospitalized in the Department of Rehabilitation Medicine of the Second Affiliated Hospital of Anhui Medical University from January 2022 to September 2022 for rehabilitation. In total, there were 10 patients with a left-sided cerebral infarction and 10 with a right-sided cerebral infarction in these 20 patients (15 males and 5 females; average age, 56.35 ± 6.81). Participants had suffered a stroke for 51.8 ± 25.5 days. (As shown in Table 1).

### 3.2. Activation of Brain Regions in Grip Tasks of Different Strengths

Box plots were used to identify changes in the HbO in the ROIs of each grip strength group, revealing no outliers. The difference between the task state and rest state in the ROI of each group followed a normal distribution (*p* > 0.05) on the Shapiro–Wilk test. In addition, we used the paired sample *t*-test to compare the mean HbO of the task state and the mean HbO of the rest state of each ROI in the BLOCK tasks. The results showed that when the force was 20% of MVC, each ROI was not activated (*p* > 0.05), while there were significant changes in the HbO in each ROI in 50% and 80% of the MVC tasks (*p* < 0.05). (As shown in Table 2 and Figure 2).

### 3.3. Difference in HbO Changes in ROIs of Different Grip Strength Groups

According to the boxplot, all groups, except for the iOL area, showed normal values. The Shapiro–Wilk test showed that all groups were normally distributed (*p* > 0.05). Levene’s test showed the homogeneity of variance for each group. Except for iOL, there were statistical differences in the HbO concentration in ROIs among the different grip strength levels (*p* < 0.05). According to a one-way ANOVA and LSD, we suggest that comparing the changes in the HbO concentration between different groups, the changes in the HbO concentration in the PFC and SMC in both hemispheres were statistically significant between the 20% and 50% MVC grip groups, as well as between the 20% and 80% MVC groups, but not between the 50% and 80% MVC groups. Moreover, the changes in the HbO concentration in the contralesional area were only statistically significant between the 20% and 80% MVC groups (*p* = 0.012). When the force was set to 50% and 80% of MVC, the changes in the HbO concentrations in all ROIs were not statistically significant. (As shown in Figure 3).

### 3.4. Lateral Index

The SMC and OL regions in the unaffected hemisphere were significantly activated in the 20%, 50%, and 80% MVC tasks. When the force was 50% of the MVC, the PFC in the contralesional hemisphere was dominant. The PFC in the affected hemisphere dominated in the 20% and 80% MVC handgrip tasks. (As shown in Figure 4.)

## 4. Discussion

In this study, we examined the changes in the HbO in the different brain ROIs of ischemic stroke patients while they performed handgrip tasks with 20%, 50%, and 80% of MVC force. After confirming that tasks will not affect each other, we believed that this order is also easy to communicate with the subjects, which is conducive to their cooperation. Similar studies [42,43] use a random sequence, but their experimental designs include different tasks in the same block paradigm. Our experiment adopts three independent block paradigms (each contains different tasks), so we believe that the order of the task execution will not affect the experimental results. Overall, we found that each ROI of the affected hemisphere was significantly activated when the grip strength reached 50% and 80% of MVC. This suggests that grip training on the contralesional side in patients with a cerebral infarction can activate the affected cerebral hemisphere. Thus, we propose that exercising the sound hand could be helpful for the functional recovery of the affected hemisphere in stroke patients. Shibuya et al. [44,45] used fNIRS to detect changes in the cerebral blood oxygenation in healthy individuals during prolonged handgrip tasks, and the results showed that the increased activation of the ipsilateral cortex had a complementary activating effect on the contralateral cortex, which contributed to the maintenance of the handgrip strength. In addition, it has also been reported [46] that the functional connectivity between the sensory-motor areas of both cerebral hemispheres increases as the HbO in the ipsilateral cortex increases when subjects perform grip strength tasks. At present, we believe there are three possible reasons for this: 1. after a cerebral infarction, the peri-infarct fibers compensate for the infarct site accordingly, resulting in increased blood oxygenation around the injury site, which facilitates neurological remodeling and enhances the functional connectivity between the peri-infarct compensated area and other brain regions [47,48]. 2. Unilateral hand-grip training, can simultaneously activate the contralesional cerebral cortex, and the ipsilesional hemisphere through the cortico-cortical pathway, which in turn can enhance the muscle strength of the untrained limb [22]. 3. Anatomically, exercising with one limb is often accompanied by the activation of the primary motor cortex in the contralateral hemisphere [49]; this is because 90% of the corticospinal tract fibers cross to the contralateral side after nerve impulses travel down through the corticospinal tract (CST) to the intersection of the medulla oblongata, while 10% of the corticospinal tract fibers cross the ipsilateral side, and 10% of corticospinal tracts in the ipsilateral hemisphere may play a key role in facilitating the recovery of patients with a severe stroke [50].

Both 50% MVC and 80% MVC can induce the activation of the ROIs in the affected and unaffected hemispheres, but in all ROIs, the concentration of HbO was higher when the force reached 80% MVC than when it reached 50% MVC. This may be related to the strength and difficulty of the task [51]. In other words, changes in the HbO concentration were positively correlated with the grip strength and task difficulty. However, the LSD test indicated that there was no statistical difference in the HbO concentration in all ROIs when the force reached 50% and 80% of MVC. This may be because 50% of MVC returns to the baseline faster than 80%, although the concentration of HbO could return to the baseline level within 15 s in all tasks, which may have a potential impact on the results. Additionally, we do not use the other fNIRS literature’s calculation method [52] that uses only the task state HbO minus the baseline value (which is 0), i.e., directly using HbOTS for intergroup comparisons. We chose ΔHbO = HbOTS − HbORS. In this study, ΔHbO was used to compare changes in the grip strength between different levels. The calculated ΔHbO values represent the task itself with respect to a specific ROI and eliminate many neurological and non-neurological factors that influence the results [53] (mentioned in the previous response), so we believe it can be compared with other tasks and other ROIs. Additionally, this might subtly affect the result. In addition, considering that patients reached the threshold of cortical excitation by performing 50% of the MVC grip task [54,55], although increasing the grip strength could lead to an increase in the blood oxygen levels, there was no qualitative change in the degree of cortical activation. This was similarly demonstrated by the calculation of LI. The LI_SMC_ was closer to zero at 50% MVC than at 80% MVC, which suggested that the iSMC played a greater role at 50% of MVC. These results indicated that moderate-intensity grip training on a sound hand can activate the affected hemisphere. In addition, some studies have demonstrated that the activation of the PFC is associated with attention as well as the execution of high-level exercises [56,57]. The LI_PFC_ in our study was negative at 50% MVC and positive at 20% and 80% MVC, which may indicate that the theory of interhemispheric inhibition does not apply to the PFC. However, the compensatory role of the PFC in patients performing high-level exercise cannot be ignored, as confirmed by the study of Jiang et al. [58]. In this study, the patients were required to gaze at the handgrip dynamometer readings to align their grip strength with the target level, and participants’ attention and vision may lead to an additional interference with the data acquisition in the PFC region of both cerebral hemispheres, which may have prevented a more accurate analysis of the bilateral PFC. Moreover, when we explored the changes in oxygenated hemoglobin in patients’ OLs, we found that when the force was set at 50% and 80% of MVC, the bilateral OLs were activated, but there was no significant difference in the iOL between the different groups. However, the concentration of HbO in the cOL between groups showed a significant difference when the force reached 80% of the MVC. Such phenomena prove that the iOL does not play a complementary interhemispheric role in grip tasks. This is related to the function of the OL itself, which is mainly responsible for functions such as motion perception and the processing of visual information [59]. Different trends in HbO changes in the PFC, SMC, and OL in grip tasks suggest that we should not focus our eyes on one brain region when assessing the interhemispheric interactions of patients, but rather consider the role between the ROIs [60], such as bilateral supplementary motor areas, cingulate motor areas, and prefrontal areas, as all have connections with the corpus callosum [61].

It has been suggested that stimulating the sound limb causes an increased interhemispheric inhibition, which could have a negative impact on the infarcted hemisphere [62]. This point has previously been argued using a bimodal equilibrium model [32]. Stroke patients with a smaller extent of brain damage and a higher degree of functional preservation in the affected hemisphere have a better recovery than those who have accompanying severe brain damage when their unaffected cerebral hemisphere was inhibited [63]. Lin [64] investigated the relationship between the Fugl–Meyer scale and the interhemispheric inhibition of stroke patients, showing that when the score was lower than 43, the significance of the interhemispheric inhibition decreased as the score decreased. When the score was higher than 43, the degree of interhemispheric inhibition increased with an increase in the score. This suggests that not all patients can promote the rehabilitation of the affected side by exercising their contralesional limb. This finding reminded us that when formulating rehabilitation treatment plans for patients, it is necessary to conduct individualized assessments of patients and determine the degree of functional preservation of the ipsilesional hemisphere. Formulating individualized rehabilitation prescriptions according to our assessments will help in the implementation of a precise rehabilitation and will be more conducive to the functional recovery of patients. When we analyzed the collected data, we found a relatively large variation in the standard deviation of the HbO values, which may be related to the state of some patients in the experiment. For example, when patients are at rest, hyperventilation could lead to a decrease in the HbO levels, resulting in a negative value due to a decrease in the circulating blood volume because of the constriction of pericortical blood vessels [65]. Therefore, there may be an increase in the standard deviation when averaging the data, but these can be statistically analyzed within the normal range. In addition, the number of infarcted foci may affect the data; for example, Patient No. 16 and No. 19 had three infarction foci, and the mean value of ΔHbO in the SMC region on the affected side at 20%, 50%, and 80% of MVC tasks was smaller than that of patients with only one infarct focus. In addition, differences in the locations of infarct sites may also have an impact on the cortical blood flow, such as the cerebellum, which can play a mediating role during the functional recovery after a basal ganglion infarction [66].

During the practical rehabilitation of stroke patients, non-invasive rehabilitation therapies, such as repetitive transcranial magnetic stimulation (rTMS) and transcranial direct current stimulation (tDCS), have become popular research topics. Although rTMS and tDCS have been reported [67,68] to reduce the inhibitory effect of the contralesional hemisphere on the affected hemisphere to promote neural remodeling on the paretic side, their large stimulation range and lack of precision and specificity would activate or inhibit all cells around the stimulated area, which may result in significant differences in treatment effects [69]. The more severe the infarction, the less likely it is that stroke patients will be able to actively participate in rehabilitation [70]. However, stroke patients must often perform active repetitive exercises to improve their neuroplasticity and thus facilitate the rehabilitation of paretic limbs [71]. Compared with the use of a dysfunctional paretic hand, patients are more willing to perform some active training with their sound hand; therefore, training patients’ unaffected limbs is not a bad way to encourage active participation in rehabilitation.

Although this study revealed changes in HbO in bilateral hemispheric ROIs in stroke patients during different grip tasks using the able-bodied hand, it is still limited by some conditions. In this experiment, we collected both 10 patients with right-sided hemiparesis and 10 patients with left-sided hemiparesis, and the fNIRS data of patients with a left-sided hemisphere infarction were flipped using a mirror transformation. Although some studies have demonstrated the feasibility of this method [30], they failed to consider whether there was a difference in the activation of the dominant or nondominant hemisphere by the limb activity. Analyzing this would require the recruitment of more subjects for a more detailed analysis. Second, there may be differences in patients performing the unaffected grip task due to a different infarct location and infarct sizes, and we will continue to collect data to further elucidate the effectiveness of cross-education in the rehabilitation for stroke patients. The present trial was only an immediate test for stroke patients with different grip training tasks on the contralesional limb, and there was no assessment of the efficacy after long-term training, which is the next step in our research. In addition, as for the non-significant results between 80% and 50%, our research group will further consult the literature to explore how to conduct a more standardized data processing. Finally, because the number of occipital channels was too low, and the collected data could not fully reflect the function of the OL, the OL was not adequately studied in this investigation. In future research, we will recruit more stroke patients, group them more carefully according to their course of disease, lesion location, and limb function, and use an fNIRS head cap equipped with a more scientific channel arrangement to further explore the role of the sound hand in promoting the recovery of the paretic limb in stroke patients.

## 5. Conclusions

In conclusion, this study demonstrated that when stroke patients with a poor upper limb function accomplished handgrip tasks with sound hands where the force reached 50% of MVC, the affected cerebral hemisphere is strongly activated. This facilitates the remodeling of the peripheral nerves of the infarct site, reduces the difficulty of implementing active rehabilitation exercises in patients’ actual clinical practice, and promotes the rehabilitation of paretic limbs.

## Figures and Tables

**Figure 1 brainsci-13-00141-f001:**
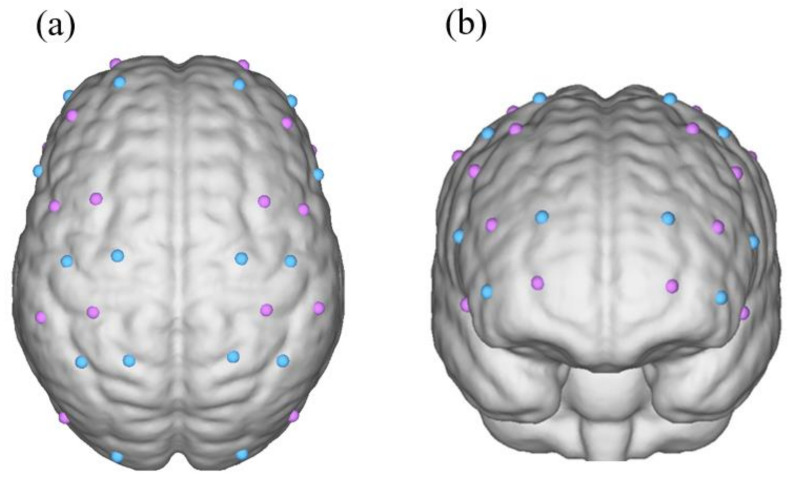
Location of the sources and detectors on the 3D model. (**a**) The vertical view of probe layout in both hemispheres; the purple point indicates light sources, the blue point indicates detectors. (**b**) The front view.

**Figure 2 brainsci-13-00141-f002:**
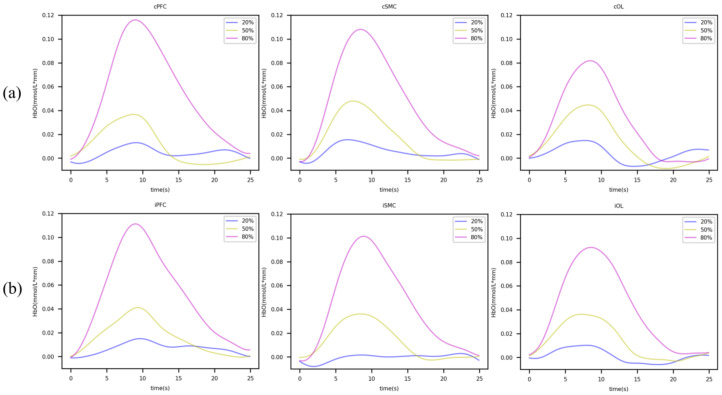
Real-time changes in HbO in the block paradigm. (**a**) Changes in HbO in the task and rest state in ROIs in the contralesional hemispheres. (**b**) Changes in HbO in task and rest state in the ipsilesional hemispheres’ ROIs. Blue, yellow, and purple lines indicate the 20%, 50%, and 80% MVC, respectively.

**Figure 3 brainsci-13-00141-f003:**
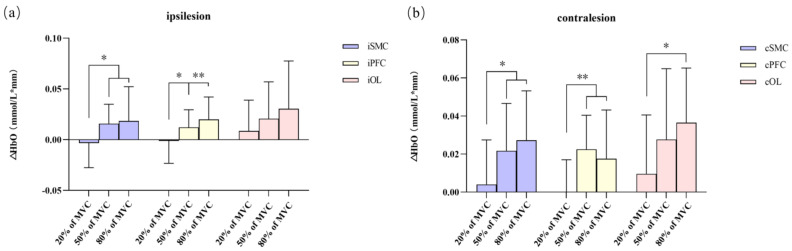
One-way ANOVA was used to detect differences in ΔHbO changes in each ROIs in the different grip strength groups. (**a**) ΔHbO changes in different ROIs of the ipsilesional hemisphere. (**b**) ΔHbO changes in different ROIs in the contralesional hemisphere. (Note: Values are mean ± SD. * 0.01 < *p* < 0.05, ** *p* < 0.01).

**Figure 4 brainsci-13-00141-f004:**
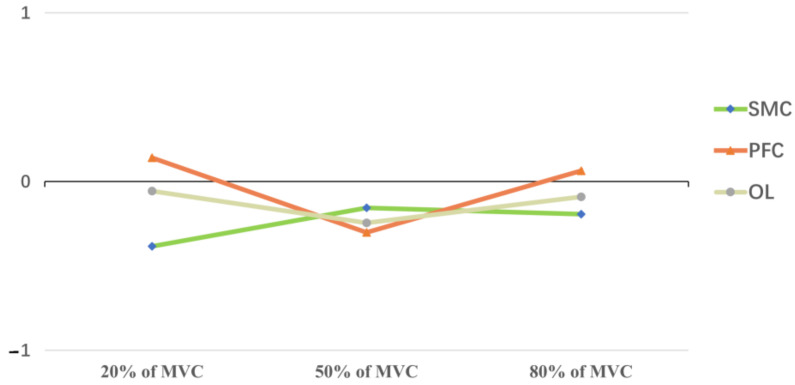
Laterality index of ROIs at different maximal voluntary contractions; with SMC at 20%, 50% and 80% of MVC, the lateralization index (LI) = −0.384, −0.156, −0.193; when PFC at 20%, 50%, and 80% of MVC, the lateralization index (LI) = 0.142, −0.302, 0.064; when OL at 20%, 50%, and 80% of MVC, the lateralization index (LI) = −0.057, −0.244, −0.090.

**Table 1 brainsci-13-00141-t001:** Clinical details of stroke patients.

Subjects Number	Sex	Age (Year)	Site of Lesion	Course of Disease (Days)	Affected Side	MMSE	MMT	MVC(Kg)	FMA
1	M	56	R, PI	31	L	30	2+	23.3	40
2	M	52	R, PI, BGI	62	L	30	2	18.5	42
3	M	50	R, PLI, TLI, BGI	35	L	28	1+	15.8	36
4	M	53	R, BGI	105	L	29	2+	18.2	39
5	M	52	R, BGI	96	L	27	1	14.5	26
6	M	58	R, BGI	38	L	28	3	21	42
7	M	53	R, PI	64	L	30	2	17.5	36
8	F	53	R, PLI, BGI	36	L	28	0	13.6	22
9	M	64	R, BGI, PI	39	L	30	0	18.4	28
10	M	59	R, BGI, PLI	58	L	26	3	22.6	35
11	M	57	L, BGI, PI	48	R	30	0	16.8	27
12	M	65	L, TI	31	R	25	0	14.8	29
13	F	64	L, BGI	112	R	30	3	13.5	33
14	M	53	L, PI	41	R	28	2+	14.8	37
15	M	40	L, BGI	55	R	30	0	22.5	21
16	F	60	L, FLI, TLI, PLI	30	R	29	0	9.9	36
17	M	51	L, FLI, PLI	33	R	30	3	19.8	40
18	M	65	L, PI	60	R	27	0	10.5	20
19	F	53	L, FLI, BGI, OLI	32	R	28	0	11.5	19
20	F	69	L, BGI	30	R	30	0	14.9	26

Abbreviations: BGI, basal ganglia infarction; F, female; FLI, frontal lobe infarction; FMA, Fugl–Meyer; L, left; M, male; MMSE, Mini-mental State Examination; MMT, manual muscle testing; MVC, maximal voluntary contraction; OLI, occipital lobe infarction; PI, paraventricular infarction; PLI, parietal lobe infarction; R, right; TLI, temporal lobe infarction; TI, thalamic infarction.

**Table 2 brainsci-13-00141-t002:** Activation of brain regions in grip tasks with different strengths.

	ROI	cSMC	iSMC	cPFC	iPFC	cOL	iOL
20% of MVC	HbOTS	0.007 ± 0.023	−0.002 ± 0.022	0.004 ± 0.017	0.006 ± 0.017	0.009 ± 0.019	0.006 ± 0.026
	HbORS	0.003 ± 0.017	0.001 ± 0.017	0.004 ± 0.016	0.007 ± 0.020	−0.000 ± 0.026	−0.002 ± 0.019
	T	0.77	−0.613	0.008	−0.232	1.393	1.265
	P	0.451	0.547	0.993	0.819	0.180	0.221
	d	0.172	0.137	0.002	0.052	0.311	0.283
50% of MVC	HbOTS	0.028 ± 0.029	0.021 ± 0.027	0.023 ± 0.027	0.022 ± 0.022	0.028 ± 0.041	0.025 ± 0.044
	HbORS	0.014 ± 0.031	0.006 ± 0.024	0.001 ± 0.024	0.010 ± 0.023	0.001 ± 0.028	0.004 ± 0.024
	T	2.374	3.706	5.603	3.105	3.320	2.557
	P	0.028 *	0.001 **	0.000 ***	0.006 **	0.004 **	0.019 *
	d	0.531	0.829	1.253	0.694	0.742	0.572
80% of MVC	HbOTS	0.061 ± 0.042	0.053 ± 0.047	0.061 ± 0.041	0.061 ± 0.047	0.049 ± 0.055	0.056 ± 0.067
	HbORS	0.034 ± 0.032	0.034 ± 0.042	0.043 ± 0.042	0.041 ± 0.040	0.013 ± 0.042	0.026 ± 0.037
	T	4.684	2.440	3.071	4.045	5.669	2.893
	P	0.000 ***	0.025 *	0.006 **	0.001 **	0.000 ***	0.009 **
	d	1.047	0.546	0.687	0.904	1.267	0.647

Note: Paired *t*-test was used to detect different changes in HbO between the task and rest states in each regions of interest (ROIs); * 0.01 < *p* < 0.05, ** 0.001 < *p* < 0.01, *** *p* < 0.001.; HbOTS, average concentration of oxygenated hemoglobin (HbO) in the task state; HbORS, average concentration of HbO in the resting state; cSMC, contralesional sensorimotor cortex; iSMC, ipsilesional sensorimotor cortex; cPFC, contralesional prefrontal cortex; iPFC, ipsilesional prefrontal cortex; cOL, contralesional occipital lobe; iOL, ipsilesional occipital lobe. According to Cohen’s *d* = 0.2 be considered a “small” effect size, 0.5 represents a “medium” effect size and 0.8 a “large” effect size.

## Data Availability

The data presented in this study are available on request from the corresponding author.

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
