# Peer review of "Cerebral Hemodynamic Changes during Unaffected Handgrip Exercises in Stroke Patients: An fNIRS Study"

_brainsci, 2023, doi:10.3390/brainsci13010141_

Round 1

Reviewer 2 Report

Where does this conclusion come from? “Overall, this research suggests that stroke patients who have poor upper limb function can select 50% of MVC grip task to do some motor training with their sound hands for the purpose of activating the ipsilesional hemisphere.”

I have a main concern about table 2. HbORS shows different values with different MVC. This could cause bias for the results ΔHbORS. It takes time for HbO to come back to baseline. Shouldn’t HbOTS just be compared with the baseline? eg. ΔHbO = HbORS – HbO_baseline

Please show the standard deviation in Figure 2.

Please show the standard deviation in Figure 4 to help understand the Lateral Index results.

In 253, There is a duplicate ‘cOL’.

Reviewer 3 Report

The authors have conducted a fNIRS study in stroke patients patients performing a hand grip task with the unafected hand at different contraction intensities (20%, 50% and 80% of MVC). They observe that 50% MVC strongly activates the affected sensoriomotor cortex, proposing this result as a potential strategy for rehabilitation in some cases of stroke.

The study is of interest, it is well written, the introduction provides a good view of the research topic and the hypothesis and objectives are adequately addressed. There are, however, some issues that the authors must check:

- Statistical results, including table 2, should provide effect size information.

- Figure 2 quality should be improved. Text font size is too small.

- Lines 252-253. cOL, contralesional occipital lobe is repeated. iOL should be written instead.

- The writting of some paragraphs should be revised. Lines 266-269: when they say "...were statistically significant in 20% and 50%..." should say " ...between 20% and 50%..."; when they saly "...as well as in 20% and 80%..." should say "... between 20% and 80%..."; when they say "...except in 50% and 80%..." they should say "...between 50% and 80%..."; When they say "... only statistically significant in 20% of MVC and 80%..." they should say "...between 20% of MVC and 80%...".

- Line 301. Instead of "nerves", perhaps "fibers" is most adequate.

- Lines 301-304. The authors describe the first possible reason for the HbO increase in ipsilateral cortex. A literature reference should be provided for such explanation. As they do for the sencond and the third ones.

Round 2

Reviewer 1 Report

The authors addressed most of my concerns. I only have two questions that would like to be further clarified.

As for major point 1, the authors claimed that the task of grasping at 80% of MVC is more difficult than that of 20% and 50%, which can cause a stronger degree of cerebral hemodynamic changes [2]. And its time to return to the baseline level does not change, so its HbORS will increased. Doesnt it mean that the baseline level of 80% did not return to a comparable level as 20% and 50%, which could cause non-significant results between 80% and 50%? Please further clarify this question and discuss it in the ms. if needed.

As for minor point 6, the authors claimed t that 10 minutes apart can eliminate the impact of the previous experiment then why say 80% of MVC as a more difficult task could impact the patients cortical blood flow, therefore, always going for 20%-50%-80% sequence? I suggest the authors also mention the order effect issues in the discussion. Although it would not affect the main conclusion, but it is important for the reader to understand the potential issues. 

Author Response

Dear Reviewer,

Thank you very much for your decision letter and additional advice on our manuscript entitled “Cerebral hemodynamic changes during unaffected hand grip exercises in stroke patients: An fNIRS study”. We also thank you for the review of our revised manuscript and further comments. We are pleased to have the opportunity to address their additional concerns. Our point-by-point responses to the latest comments are listed below this letter. 

Look forward to hearing from you soon. 

    With best wishes, 

                                                                                                        Yours sincerely, 
                                                                                                                 Yuqin Ma

Specific Comments

Q1: As for major point 1, the authors claimed that “the task of grasping at 80% of MVC is more difficult than that of 20% and 50%, which can cause a stronger degree of cerebral hemodynamic changes [2]. And its time to return to the baseline level does not change, so its HbORS will increased.” Doesn’t it mean that the baseline level of 80% did not return to a comparable level as 20% and 50%, which could cause non-significant results between 80% and 50%? Please further clarify this question and discuss it in the ms. if needed.

Response: Thank you for your insightful suggestion. I am sorry that the last response was not very clear about the interpretation of the baseline. We all know that the cerebral hemodynamic change measured by fNIRS is a relative value. And during the data processing, we chose [-2 s,0 s] HbO before the start of the task as the baseline [1]. So the baseline is the same in different grip tasks. In addition, we can see from Fig.2 that the concentration of HbO could return to the baseline level within 15 s in all tasks (as shown in Fig.2), so the baseline level of 80% returned to a comparable level as 20% and 50%. As for the non-significant results between 80% and 50%, we thought, perhaps, this was because 20% and 50% get back to baseline faster than 80%, which may have potential impact on results. In addition, we do not use other fNIRS literature’s calculation method [3] that using only the task state HbO minus the baseline value (which is 0) i.e. directly using HbOTS for intergroup comparisons. We chose ΔHbO = HbOTS - HbORS. In this study, ΔHbO was used to compare changes in grip strength between different levels. The calculated ΔHbO values represent the task itself with respect to a specific ROI and eliminate many neurological and non-neurological factors that influence the results [2] (mentioned in the previous response), so we believe it can be compared with other tasks and other ROIs. And this might subtly affect the result. Therefore, our research group will further consult literature to explore how to conduct more standardized data processing. What’s more, we will also explain this part in our manuscript. Thank you for your invaluable suggestions again! (Line 403-413, 504-506)

[1] Wu YJ, Hou X, Peng C, et al. Rapid learning of a phonemic discrimination in the first hours of life. Nat Hum Behav. 2022;6(8):1169-1179. doi:10.1038/s41562-022-01355-1

[2] Tachtsidis I, Scholkmann F. False positives and false negatives in functional near-infrared spectroscopy: issues, challenges, and the way forward [published correction appears in Neurophotonics. 2016 Jul;3(3):039801]. Neurophotonics. 2016;3(3):031405. doi:10.1117/1.NPh.3.3.031405

[3] Ge, R., Wang, Z., Yuan, X., Li, Q., et al. The effects of two game interaction modes on cortical activation in subjects of different ages: A functional near-infrared spectroscopy study. IEEE Access, (2021)9:11405-11415.

Q2: As for minor point 6, the authors claimed t that 10 minutes apart can eliminate the impact of the previous experiment then why say 80% of MVC as a more difficult task could impact the patient’s cortical blood flow, therefore, always going for 20%-50%-80% sequence? I suggest the authors also mention the order effect issues in the discussion. Although it would not affect the main conclusion, but it is important for the reader to understand the potential issues.

Response: Thank you for your kindly reminder. After confirming that tasks will not affect each other, we adopted the inertial thinking from easy to difficult. At the same time, we believed that this order is also easy to communicate with the subjects, which is conducive to their cooperation. Similar studies [1, 2] use a random sequence, but their experimental designs include different tasks in the same block paradigm. Our experiment adopts three independent block paradigms (Each contains different tasks), so we believe that the order of task execution will not affect the experimental results. But your suggestion is very insightful. We are glad to mention the order effect issues in our manuscript. Thank you very much! (line 366-371)

[1] Du Q, Luo J, Cheng Q, Wang Y, Guo S. Vibrotactile enhancement in hand rehabilitation has a reinforcing effect on sensorimotor brain activities. Front Neurosci. 2022;16:935827. Published 2022 Oct 4. doi:10.3389/fnins.2022.935827

[2]Yang CL, Lim SB, Peters S, Eng JJ. Cortical Activation During Shoulder and Finger Movements in Healthy Adults: A Functional Near-Infrared Spectroscopy (fNIRS) Study. Front Hum Neurosci. 2020;14:260. Published 2020 Jul 8. doi:10.3389/fnhum.2020.00260
